# Divergent confidence intervals among pre-specified analyses in the HiSTORIC stepped wedge trial: An exploratory post-hoc investigation

**Richard A. Parker**[1]*, **Catriona Keerie**[1], **Christopher J. Weir**[1], **Atul Anand**[2], **Nicholas L. Mills**[2,3]

1 Edinburgh Clinical Trials Unit, Usher Institute, The University of Edinburgh, Edinburgh, United Kingdom, 2 British Heart Foundation/University Centre for Cardiovascular Science, The University of Edinburgh, Edinburgh, United Kingdom, 3 Usher Institute, The University of Edinburgh, Edinburgh, United Kingdom

* Richard.Parker@ed.ac.uk

**Data Availability Statement:** Patients did not individually consent to the HISTORIC trial. Raw datasets were only made available via restricted access within a secure research safe haven

## Abstract

### Background

The high-sensitivity cardiac troponin on presentation to rule out myocardial infarction (HiSTORIC) study was a stepped-wedge cluster randomised trial with long before-and-after periods, involving seven hospitals across Scotland. Results were divergent for the binary safety endpoint (type 1 or type 4b myocardial infarction or cardiac death) across certain pre-specified analyses, which warranted further investigation. In particular, the calendar-matched analysis produced an odds ratio in the opposite direction to the primary logistic mixed-effects model analysis.

### Methods

Several post-hoc statistical models were fitted to each of the co-primary outcomes of length of hospital stay and safety events, which included adjusting for exposure time, incorporating splines, and fitting a random time effect. We improved control of patient characteristics over time by adjusting for multiple additional covariates using different methods: direct inclusion, regression adjustment for propensity score, and weighting. A data augmentation approach was also conducted aiming to reduce the effect of sparse data bias. Finally, the raw data was examined.

### Results

The new statistical models confirmed the results of the pre-specified trial analysis. In particular, the observed divergence between the calendar-matched and other analyses remained, even after performing the covariate adjustment methods, and after using data augmentation. Divergence was particularly acute for the safety endpoint, which had an event rate of 0.36% overall. Examining the raw data was particularly helpful to assess the sensitivity of the results to small changes in event rates and identify patterns in the data.

environment for approved individuals who had undertaken the necessary governance training. Exporting the raw datasets used in the HiSTORIC trial is prohibited under patient confidentiality and data protection regulations because it contains a large amount of potentially identifying patient information. These datasets cannot be made publically available. However, researchers can apply for access to these datasets through applying to Dataloch (https://dataloch.org/).

**Funding:** The HiSTORIC trial was funded by the British Heart Foundation (grant PG/15/51/31596) with support from British Heart Foundation Research Excellence Awards (awards RE/18/5/34216 and RE/18/6134217). NM is supported by the Butler Senior Clinical Research Fellowship and a Program Grant (fellowship FS/16/14/32023 and grant RG/20/10/34966). The funders had no role in study design, data collection and analysis, decision to publish, or preparation of the manuscript.

**Competing interests:** I have read the journal's policy and the authors of this manuscript have the following competing interests: AA and NM have received honoraria from Abbott Diagnostics. In addition, NM reports research grants awarded to the University of Edinburgh from Abbott Diagnostics and Siemens Healthineers outside the submitted work and honoraria from Siemens Healthineers, Roche Diagnostics, and LumiraDx. The other authors have no competing interests to declare.

## Conclusions

Our experience reveals the importance of conducting multiple pre-specified sensitivity analyses and examining the raw data, particularly for stepped wedge trials with low event rates or with a small number of sites. Before-and-after analytical approaches that adjust for differences in patient populations but avoid direct modelling of the time trend should be considered in future stepped wedge trials with similar designs.

## Introduction

The stepped wedge cluster randomised controlled trial design has seen a surge in popularity over the last twenty years, particularly since 2010 [1]. In this design, all clusters begin in the control condition and then cross over to the intervention condition at pre-specified time points [1, 2]. Instead of clusters being randomised to groups as in standard cluster randomised trials, each cluster is randomised to a particular crossover time point.

The HiSTORIC trial was a prospective stepped-wedge cluster randomised controlled trial involving seven tertiary and secondary care hospitals across Scotland [3]. The stepped wedge trial design incorporated long before and after periods, which compressed the period of intervention rollout. This was because prolonged parallel running of different clinical pathways between different hospitals within the same health board was felt to be a clinical risk as staff worked across multiple hospitals within the board.

The aim of the trial was to evaluate the effectiveness and safety of implementation of an intervention for ruling out myocardial infarction in consecutive patients presenting at an emergency department with suspected acute coronary syndrome. Full details of the trial design and results are reported elsewhere [3].

The intervention involved implementation at the site level of an early rule-out pathway for patients with suspected acute coronary syndrome [4]. Under this intervention pathway, patients with high-sensitivity cardiac troponin I concentrations <5 ng/L at presentation are considered low risk and myocardial infarction is ruled out without further testing, unless they present early with symptom onset <2 hours from presentation where cardiac troponin is retested 3 hours after presentation. Patients with cardiac troponin concentrations ≥5 ng/L at presentation but below the sex-specific 99th centile are retested 3 hours after presentation. Myocardial infarction is ruled out at 3 hours if cardiac troponin concentrations are unchanged (<3 ng/L change) and remain below the 99th centile diagnostic threshold for myocardial infarction on retesting.

The co-primary outcomes were length of hospital stay (efficacy endpoint) and the proportion of patients with type 1 or type 4b myocardial infarction or cardiac death after discharge and within 30 days of index presentation (safety endpoint). As reported in Anand et al. [3], there were 31,492 patients enrolled across seven sites, with a mean age of 59 (standard deviation [SD] 17) and 45% were women. Length of stay was significantly reduced from 10.1 to 6.8 hours following implementation of the intervention into practice with a geometric mean ratio of 0.78 (95% confidence interval [CI] 0.73 to 0.83, P<0.001) [3]. However, non-inferiority was not demonstrated for the 30-day safety endpoint: the upper limit of the one-sided 95% confidence interval for the adjusted approximate risk difference was 0.70% compared to a non-inferiority margin of 0.50%, even though the observed proportion of safety outcome events was lower in the intervention period compared to the control period (0.3% *versus* 0.4%). Note that

there were just 113 patients (0.36%) in whom a safety outcome event was recorded over the entire 24-month study period, indicating sparse data.

The safety endpoint findings were strongly dependent on the model fitted and were inconsistent across sensitivity analyses [3]. In particular, the odds ratio for the primary analysis was 1.97 (95% CI 0.95 to 4.08) but for the calendar matched analysis it was 0.48 (95% CI 0.29 to 0.80). Not only was the point estimate for the odds ratio in the opposite direction for the calendar matched analysis, but the 95% confidence intervals were non-overlapping. Expressed as an absolute risk difference, the point estimate was 0.26% for the primary analysis compared with -0.26% for the calendar matched analysis. The "as randomised analysis", where the analysis was based on the planned rather than the actual dates of implementation, gave an odds ratio of 0.53 (95% CI 0.23 to 1.22).

In contrast, the length of stay outcome results were more consistent across analyses. For example, the primary analysis had a geometric mean ratio of 0.78 (95% CI 0.73 to 0.83) compared to 0.65 (95% CI 0.62 to 0.68) for the calendar-matched, although note that confidence intervals were still non-overlapping. A significant effect was observed on the efficacy endpoint across all analyses except for the "as-randomised" analysis (geometric mean ratio 1.00, 95% CI 0.93 to 1.07). Note that since the actual dates of implementation were different from the "as-randomised" cross-over time points, the value of the "as-randomised" analysis is very limited.

Although the results of the efficacy endpoint (continuous primary outcome variable) were generally robust to sensitivity analyses, the binary primary safety endpoint results were not, and results differed greatly depending on which of the pre-specified statistical analysis methods were used. Whilst the secondary safety outcome at one year was reassuring, occurring in 2.7% and 1.8% of patients before and after implementation (adjusted odds ratio, 1.02 [95% CI, 0.74–1.40]; P = 0.894), divergence in the models used to evaluate the primary safety endpoint made interpretation of the safety outcome at 30 days challenging.

Moreover, there was a concern that small changes in the demographics or pre-existing conditions of the patient population over time could have introduced bias. Further post-hoc analyses of the HiSTORIC data were therefore required to eliminate concerns regarding this issue.

The aims of this methodological study were therefore to (i) investigate the reason (or reasons) for the divergent results, and (ii) adjust for any changes in patient demographics and baseline confounders over time.

## Methods

### The HISTORIC trial design

HiSTORIC was a stepped-wedge cluster randomized controlled trial enrolling consecutive patients with suspected acute coronary syndromes across 7 acute secondary or tertiary care hospitals in Scotland [3]. Cluster randomization of hospital sites was necessary to avoid the risk of clinical error attributable to simultaneous use of different diagnostic pathways. The trial was registered (www.clinicaltrials.gov; Unique identifier: NCT03005158), approved by the Scotland A Research Ethics Committee (REC number 12-SS-0115) and the conduct of the trial was periodically reviewed by an independent trial steering committee. We received approval to use routinely collected National Health Service (NHS) data for research without the need for individual patient consent. This approval was granted by the NHS National Services Scotland Privacy Advisory Committee (PAC), now called the NHS Scotland Public Benefit and Privacy Panel for Health and Social Care (NHSS HSC-PBPP). All data were collected from the patient record and national registries, de-identified, and linked in a data repository within secure National Health Service safe havens.

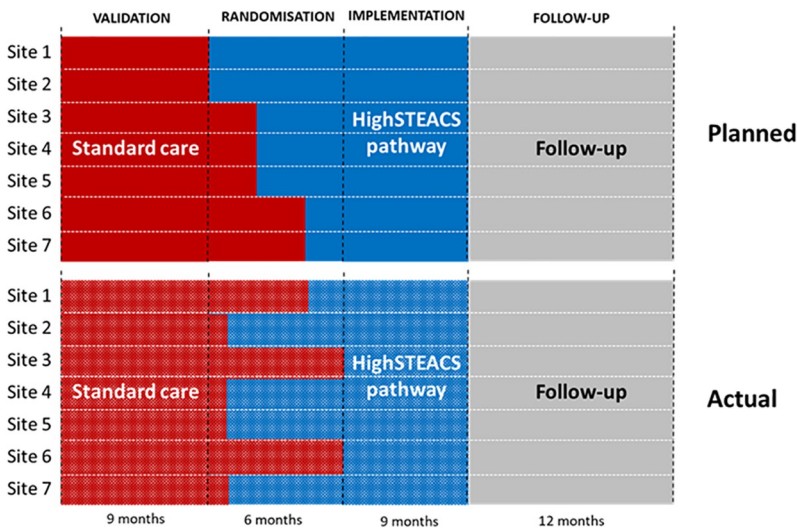

**Fig 1. Schematic diagram showing planned and actual implementation of the intervention (the HighSTEACS pathway).**

The trial design is illustrated in Fig 1. The trial consisted of three phases over a 24-month period—labelled "validation", "randomisation" and "implementation". Patients presenting during the validation phase were managed according to the established assessment protocol (standard care). This was followed by a 6-month randomisation phase where participating centres were randomised to cross over to use the early rule out pathway (intervention), with stratification by hospital size. However, the sites did not (or were unable to) adhere to the pre-planned randomised crossover time points (see comparison between "Planned" and "Actual" in Fig 1). As is common in pragmatic trials evaluating changes in care pathways, the actual date of implementation of the intervention differed from the planned date for clinical reasons that were out with the control of the investigators. These included a directive from one Health Board to implement earlier than planned to address capacity issues in the Emergency Department, and delays at other sites due to a change in the provision of ambulatory care services and a request for an additional governance review prior to implementation. A final "implementation phase" followed in which all sites used the early rule out pathway. Recruitment was continuous with patients experiencing either the standard care pathway (control) or early rule out pathway (intervention), but not both. Within the typology schema of Copas et al. [5], this trial has a "continuous recruitment with short exposure" design. Patients were included in the study on their first attendance with those who reattended during the study period not enrolled twice.

As a stepped wedge trial, the study design allowed both (i) horizontal before-and-after comparisons of outcomes (i.e. comparisons within rows/sites in Fig 1), and (ii) vertical cross-sectional comparisons based on contemporaneous controls (i.e. comparisons across sites for patients presenting at the same or similar times). Hereafter for simplicity, we refer to these as "horizontal" and "vertical" comparisons, respectively.

In this study, the clusters were seven individual sites, which were randomized to begin the intervention at three distinct time points. In the actual study design (Fig 1), although all sites implemented the early rule-out pathway during the randomisation phase, four of the sites crossed over in the first cross over point, one in the second, and two in the final cross over point.

We have previously described this trial as a "hybrid before-and-after and stepped wedge design" [6].

The HISTORIC trial used routinely collected data to identify, characterise and follow up all patients presenting at the Emergency Department and record study outcomes. Patients did not individually consent to participate in the HISTORIC trial. The primary and secondary safety outcomes were adjudicated as previously described [3]. Source data sets were linked together within a secure Safe Haven environment (the Public Health Scotland National Safe Haven [7]), and release of outputs was tightly controlled to prevent release of sensitive or confidential patient-level information. Exporting raw data or details about specific events was prohibited under patient confidentiality and data protection regulations in case this identified individual patients. Raw datasets were only accessible by approved individuals who had undertaken the necessary governance training.

## Pre-specified statistical analysis methods

A linear mixed-effects regression model was fitted to the efficacy co-primary outcome of length of hospital stay, adjusting for hospital site, season, time of patient presentation since start of study (in days), and an indicator variable for whether the new intervention pathway was implemented or not. The reason for adjusting for season in the model is that there is known to be an increased risk of myocardial infarction (and death from myocardial infarction) in the winter months [8]. Season featured in the model as a four category factor variable, with three indicator dummy variables for Spring (March April May), Summer (June July August) and Autumn (Sept Oct Nov); relative to the reference category Winter (Dec Jan Feb). Time of patient presentation since start of study appeared as a linear term in the model, centred on the mid-point of the study enrolment period. Hospital site was included as a random effect. A compound symmetry variance-covariance structure was assumed.

A logistic mixed-effects regression model was fitted to the safety co-primary outcome, using an adaptive Gauss-Hermite quadrature method to approximate the log-likelihood [9]. As for the efficacy outcome, this was adjusted for hospital site (as a random effect), season, time of patient presentation since start of study (in days), and an indicator variable for the intervention.

In the pre-specified primary analysis, data were analysed according to the actual timing of the introduction of the new pathway rather than the randomised time the new pathway was to be introduced. Pre-specified sensitivity analyses involved various modifications of the primary analysis model and analysis population, details of which are presented in Table 1. Additional post-hoc sensitivity analyses were used to assist interpretation of the findings from the pre-specified analyses.

## Statistical analysis methods

Table 1 describes the different pre-specified models fitted, as per the original statistical analysis plan. A key pre-specified secondary analysis (analysis B in Table 1) involved comparing the validation and implementation phases only of the trial in a before-and-after study context, where the validation and implementation phases were calendar matched to avoid any bias due to a seasonality effect. The rationale for this before-and-after analysis was that it enabled us to compare the intervention once it had been fully embedded into the Emergency Department with standard care. It was anticipated that during the randomisation phase of the trial the effect of the intervention may have been diluted since clinicians might still have been learning to adopt the new procedure. However, this analysis assumes that there were no natural changes in the safety outcome over the evaluation period, which is a reasonable assumption given the

**Table 1. Summary of all pre-specified primary and secondary analysis models.**

| | Analysis model | Details | Justification |
|---|---|---|---|
| A | **Primary analysis** | Mixed effects model adjusting for hospital site (as random effect), season, time of patient presentation since start of study (in days). | Standard mixed effects model using all the data. Adjusts for the time trend and seasonal effects. |
| B | **Comparison of calendar matched validation and implementation phases only** | The validation and implementation phases were calendar matched to avoid any bias due to a seasonality effect. Only presentations between 3$^{rd}$ March and 3$^{rd}$ Sept were included. The statistical model fitted was the same as the primary model except there was no model-based adjustment for time or season. | Enables a comparison of the intervention when it is fully embedded into the Emergency Department with standard care; eliminates any confounding bias due to seasonal effects; and avoids difficulties with estimating the time trend due to the long before and after periods, at the expense of the intervention comparison being potentially confounded by changes over time. |
| C | **Intervention effect allowed to randomly vary with site** | The same primary analysis model was fitted, but additionally including a random coefficient for the intervention effect | Recommended by Thompson et al. [10]. If the intervention effect varies across sites, then this may lead to biased effect estimates and poor confidence interval coverage if the standard model is fitted [10]. |
| D | **Randomisation phase only** | The same primary analysis model was fitted, but only using data collected within the randomisation phase (i.e. the period of time when there were sites in both intervention and control conditions) and based on the actual times the intervention was introduced. | This analysis reduces the risk of confounding bias due to secular changes over time (Davey et al. [11]) since the analysis is conducted over a restricted time period and vertical comparisons can be made. |
| E | **As randomised analysis** | The same primary analysis model was fitted, but using the randomised times that the intervention was supposed to have been introduced rather than the actual times. | Recommended by Hemming et al. [2] and consistent with the "Intention-to-treat" principle as applied to the sites. Can assess if the analysis results are sensitive to the timing of the actual intervention start date. |

study population primarily consists of patients who have myocardial infarction ruled out and receive few if any specific treatments.

For post-hoc analyses, our model choices were informed by the stepped wedge literature [10–15] and with consideration given to our chosen study design (see Table 2). These included using different ways of modelling the time trend (e.g. as cubic spline or categorical variable) and adjusting for the length of time that the site had been using the intervention pathway (exposure time). Note that for all post-hoc models of the binary safety endpoint we used the Laplace approximation to increase the speed of convergence.

In addition to the above models, there was a need to control for any natural changes in key demographic variables across the duration of the study. This was at least partly motivated by a peer reviewer of the primary trial report who suggested that adjusting for potential confounders through using *"propensity scores either through matching or weighting would strengthen the study substantially."*

Therefore, in all pre-specified and post-hoc models, we performed covariate adjustment to attempt to control for any shift in patient characteristics over time, and we did this using four alternative methods of adjustment listed in Table 3.

## Data augmentation

The event rate for the binary safety outcome was only 0.36% overall, and so there was a concern that sparse data bias may have contributed to the divergent results. We therefore considered methods to address this problem. One of the methods considered was a data augmentation approach as suggested by Greenland et al. [21]. We adapted this method for the context of stepped wedge trials by first generating synthetic data to encode our prior information, and then combining this synthetic data with the real HISTORIC data. For generating the prior data, we fixed the number of events before and after the cross over point in each site (and across all sites) to be equal, and the proportion of events was simulated based on a Bernoulli

**Table 2. Summary of all post-hoc advanced primary and secondary analysis models.**

| | Analysis model | Details | Justification |
|---|---|---|---|
| F | **Random slope for time: Time effect allowed to randomly vary with site** | The same primary analysis mixed effects model was fitted, but additionally including a random coefficient for the time trend. | Recommended by Thompson et al. [10]. Thompson et al. found that a random period mixed effects model was the most robust to departures from the constant intervention and period effect assumptions [10]. Also see Li et al. [12] |
| G | **Adjust for exposure time** | The same primary analysis model was fitted, but additionally adjusting for the length of time since the intervention was introduced in each site (exposure time) as a linear coefficient. | Nickless et al. [13] suggest that adjusting for exposure time should be considered in stepped wedge analyses. |
| H | **Cubic spline to model time trend** | The same primary analysis model was fitted, but instead of using a linear term to model the time trend we allowed a more flexible relationship between time and outcome by means of a cubic spline. | Davey et al have suggested adjusting for time trends via fitting a cubic spline function [11]. |
| I | **Cubic spline used to model both the time trend and exposure time** | Same as above (model H) but with the addition of adjusting for exposure time using a cubic spline. | Davey et al have suggested adjusting for time trends via fitting a cubic spline function [11], and Nickless et al. [13] suggest that adjusting for exposure time should be considered in stepped wedge analyses |
| J | **Adjusting for categorical time (3-monthly periods) and no seasonal effect** | The same primary analysis model was fitted, but instead of using a linear term to model the time trend we adjusted for categorical time, and we removed the seasonal effect terms. | Although we fitted the time trend as linear in our primary analysis to increase the degrees of freedom, the traditional model is to adjust for time as categorical (e.g. see Hussey and Hughes [14] and Hemming et al. [15]). The advantage is that we are no longer assuming a fixed linear relationship for the time effect. |
| K | **Adjusting for categorical time and exposure time as spline function** | A combination of models I and J. | Same as above, but fitting a smooth cubic spline function to model exposure time (Nickless et al. [13]) |

distribution with pre-specified probability of 0.4%, which was the event rate assumed in the original sample size calculations. Within each site, the dates of presentation (days since start of study) were fixed to be one day before and after the crossover point. The rationale for choosing these dates of presentation is that prior research suggests that cluster-periods closest to the treatment switch date are most informative [22, 23] and therefore we conjecture that the effect of sparse data would be more acute at the crossover points. We also found it easier to generate the necessary priors using these dates of presentation. The amount of prior information (i.e. number of simulated patients) was calibrated to produce an absolute risk difference of approximately 0%, and 95% CI within the bounds of -2% to 2%. This means that we are hypothetically assuming *a priori* that it is very unlikely that the absolute risk difference would exceed 2% in either direction. This prior information was then combined with the real data to observe the effect on the intervention effect estimates. We then fitted the primary analysis model (a generalised linear mixed effect model utilizing an adaptive Gauss-Hermite approximation to the log-likelihood) without intercept term, adjusting for the type of data (i.e. we specified a binary factor variable called "synthetic" taking the value 1 for synthetic data and 0 otherwise). A dramatic change in the results compared to the original analysis would have indicated the presence of sparse data bias.

## Examination of the raw data

As a post-hoc analysis, the raw safety endpoint data was examined within the secure Safe Haven analysis platform. This is consistent with the recommendation of Greenland et al. [21] to perform "tabular examination" of basic data to identify sparse data problems. Safety outcome data was categorised into monthly periods per site, called "cluster periods", and the data shown in a table. We also tested the sensitivity of the results to small changes in the numbers of events per cluster-period.

**Table 3. Summary of adjustment methods with justification.**

| | Method of adjustment | Details | Justification |
|---|---|---|---|
| 1 | **Regression adjustment for key demographic variables** | Age, sex and Scottish Index of Multiple deprivation (SIMD) were directly entered into the models as covariates. | Adjustment for a limited number covariates should not adversely affect statistical power for the binary safety endpoint analysis, whereas adjustment for a long list of variables may do. This method also avoids any possible bias in the use of propensity score methods in this context. |
| 2 | **Regression adjustment for multiple covariates** | A long list of medical history and demographic variables were directly entered into the models as covariates: prior ischaemic heart disease, myocardial infarction, cerebrovascular disease, diabetes, percutaneous coronary intervention, coronary artery bypass grafting, prior treatment with aspirin, lipid lowering drugs, beta blockers, ACE inhibitors, age, sex, and SIMD quintile. | Adjustment for prior disease markers in addition to demographic variables is likely to lead to improved control of confounders over the study duration. |
| 3 | **Regression adjustment for propensity score** | Logistic mixed regression models were fitted to the group membership variable (intervention/control) with the following covariates contributing to the propensity score estimation: prior ischaemic heart disease, myocardial infarction, cerebrovascular disease, diabetes, percutaneous coronary intervention, coronary artery bypass grafting, prior treatment with aspirin, lipid lowering drugs, beta blockers, ACE inhibitors, age, sex, and SIMD quintile. The logit of the propensity score was then included as a single linear explanatory variable in the statistical models to adjust for propensity score. Note that propensity score estimation was performed separately for the calendar matched analysis because this was based on a reduced population. | This method has been used previously in observational studies [16, 17] and has the advantage of summarising a long list of covariates as a single score and obviating the need to adjust for a long list of confounders in regression modelling; which is especially problematic for the binary safety outcome due to sparse data in this context. Although this method has been previously criticised for being biased [18, 19], in our context intervention assignment was independent of the patient baseline variables and therefore we expect any bias to be negative (towards the null) [18]. Further research may be needed however relating to the most appropriate way to model propensity score, and the consistency and unbiasedness of this method when used in the context of stepped wedge trials and/or before-after designs. |
| 4 | **Inverse probability of treatment weighting using the propensity score** | Using the same propensity scores as above, instead of adjusting for the logit of propensity score, we weighted the mixed models according to the probability of treatment actually received. Specifically, we calculated weights as: $w_i = \frac{Z_i}{p_i} + \frac{(1-Z_i)}{(1-p_i)}$, where Z = 1 if the patient is recruited under in the intervention condition and Z = 0 otherwise, and $p_i$ indicates the propensity score. This inverse probability of treatment weighting (IPTW) method is expected to yield unbiased estimates of average treatment effects [20]. | This method shares the same advantage of summarising a long list of confounders into a single score as the "regression adjustment for propensity score" method. Again, further research may be needed to evaluate the performance of this method in stepped wedge trials with a low number of sites and/or long before-and-after periods. |

As indicated in Table 1, Nickless et al. [13] recommend adjusting for exposure-time in stepped wedge analyses. This was a particular concern in our trial because the long before and after periods made it difficult to precisely estimate the intervention effect independently of any natural overall time trends or local exposure time effects using the standard model. We therefore also visually examined the raw data for any obvious exposure-time effects.

## Software

R software version 3.6.1 was used to conduct all analyses [24]. Statistical models were fitted using the lme4 package [9] and forest plots constructed using the metafor package [25].

## Results

For the length of stay outcome, results were consistent across all pre-specified and post-hoc analyses (Fig 2A and 2B). Only the "as-randomised" analysis showed a clear difference, in line with its expected bias towards the null. Fig 2A shows that adjustment for covariates produced very little change in the intervention effect estimates and confidence intervals compared to the pre-specified model results reported in Anand et al. [3] Furthermore, there were negligible differences between results due to the method of covariate adjustment.

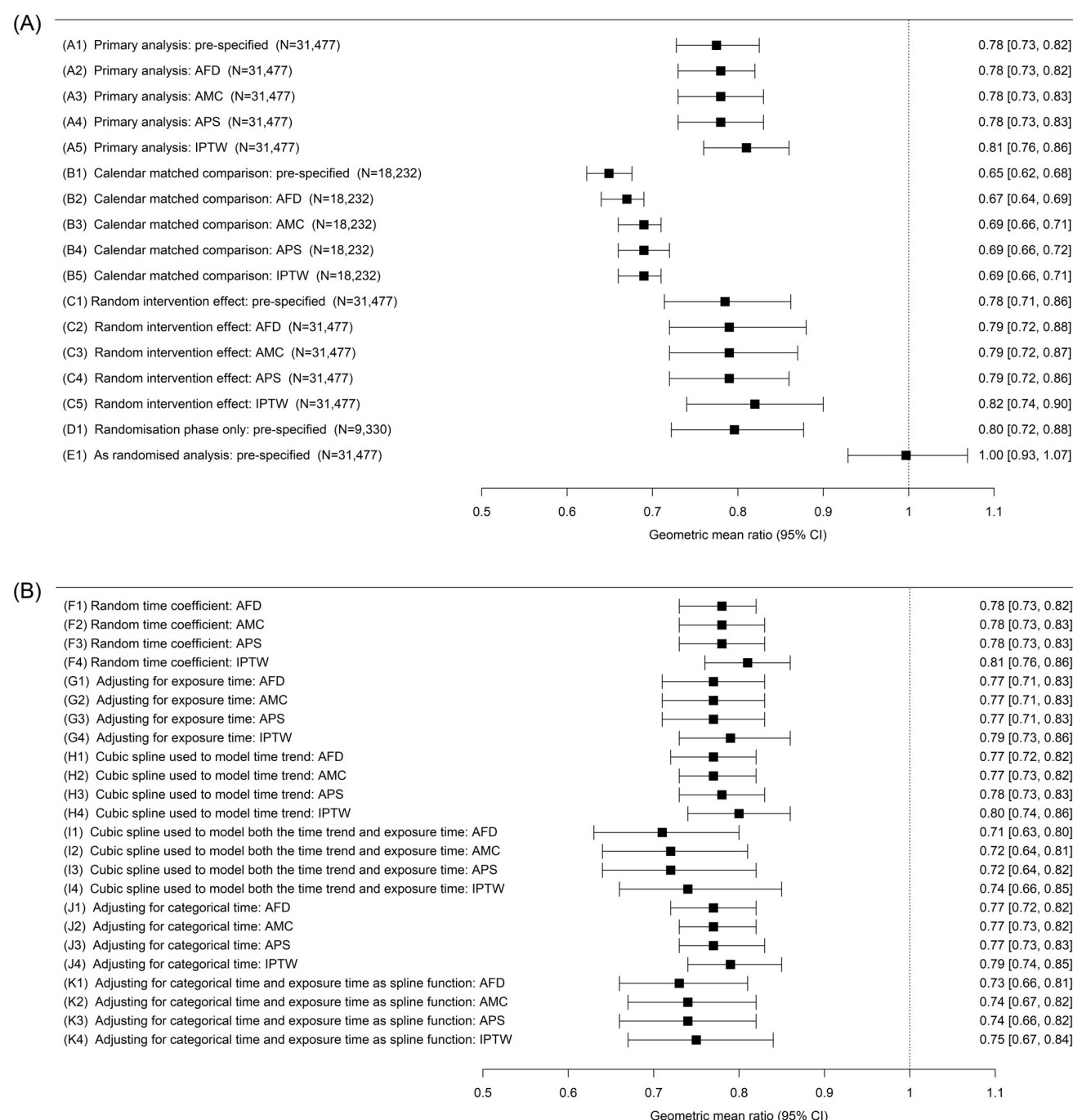

**Fig 2. A**: A forest plot showing pre-specified model results (geometric mean ratios and 95% confidence intervals) from the length of stay efficacy outcome analysis (including modifications to adjust for potential confounders). AFD = Adjusted for demographics, AMC = Adjusted for multiple covariates, APS = Adjusted for propensity score, IPTW = Inverse probability of treatment weighting. **B**: A forest plot showing advanced post-hoc model results (geometric mean ratios and 95% confidence intervals) from the length of stay efficacy outcome analysis (N = 31,477 for all). AFD = Adjusted for demographics, AMC = Adjusted for multiple covariates, APS = Adjusted for propensity score, IPTW = Inverse probability of treatment weighting.

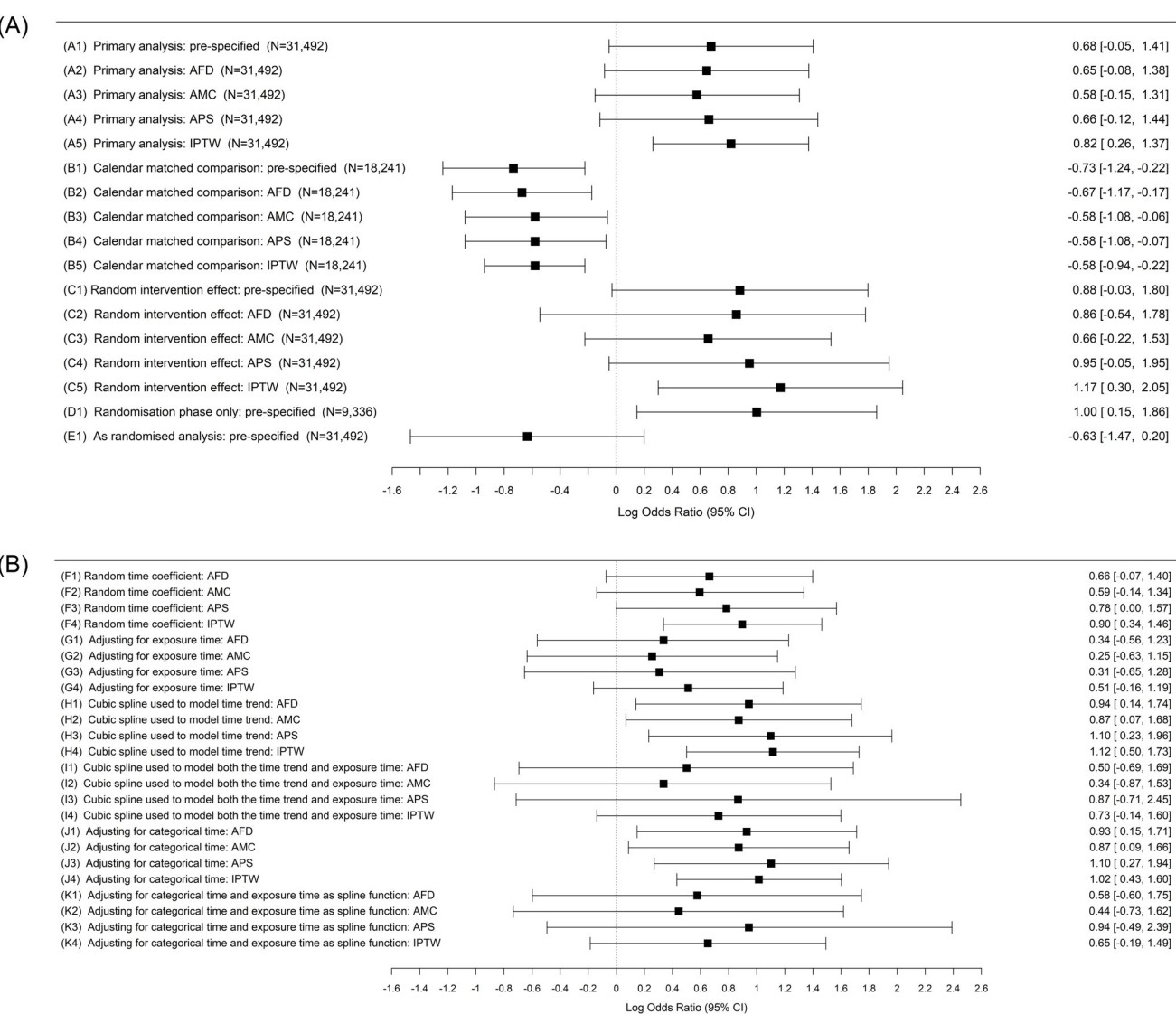

**Fig 3. A**: A forest plot showing log-odds ratio results and 95% confidence intervals from the pre-specified primary safety outcome analysis (including modifications to adjust for potential confounders). AFD = Adjusted for demographics, AMC = Adjusted for multiple covariates, APS = Adjusted for propensity score, IPTW = Inverse probability of treatment weighting. **B**: A forest plot showing advanced post-hoc model results (log-odds ratios and 95% confidence intervals) from the primary safety outcome analysis (N = 31,492 for all). AFD = Adjusted for demographics, AMC = Adjusted for multiple covariates, APS = Adjusted for propensity score, IPTW = Inverse probability of treatment weighting.

The safety endpoint results showed a similar pattern of consistency with the pre-specified findings (see Fig 3A). The calendar matched analysis results were substantially different from the primary analysis results, and these major differences remained after covariate adjustment, regardless of the method used to adjust for baseline characteristics. The advanced model results (Fig 3B) were fairly consistent for the safety endpoint in so far as the confidence intervals were all overlapping, although the odds ratio point estimate still varied from 1.29 (analysis G2: adjustment for exposure time and multiple covariates) up to 3.23 (analysis C5: IPTW with random intervention effect).

Note that the "calendar matched" and "randomisation phase only" analyses were based on a reduced sample size as shown in Figs 2A and 3A.

### Data augmentation

After performing data augmentation, the odds ratios for the primary analysis model reduced slightly from 1.97 (95% CI 0.95 to 4.08) down to 1.76 (95% CI 0.96 to 3.25), based on a synthetic prior data with a sample size of N = 1274 imaginary patients and risk difference (RD) of 0% (95% CI -0.71% to 1.99%). For a highly specific prior of N = 3220 (RD 1, 95% CI -0.45% to 0.94%), the odds ratio reduced to 1.56 (95% CI 0.92 to 2.64). Note that the number of sites and length of the randomisation phase remained unchanged in this analysis.

### Examination of the raw data

Fig 4, which was presented in the supplementary material of Anand et al. [3] and reproduced here, shows how the binary safety outcome changed over the duration of the trial.

The safety outcome event rates were 0.5% (49/10724), 0.4% (37/9336), and 0.2% (27/11432) for the first, second and third phases of the trial respectively, corresponding to the validation, randomisation, and implementation phases of the trial, respectively.

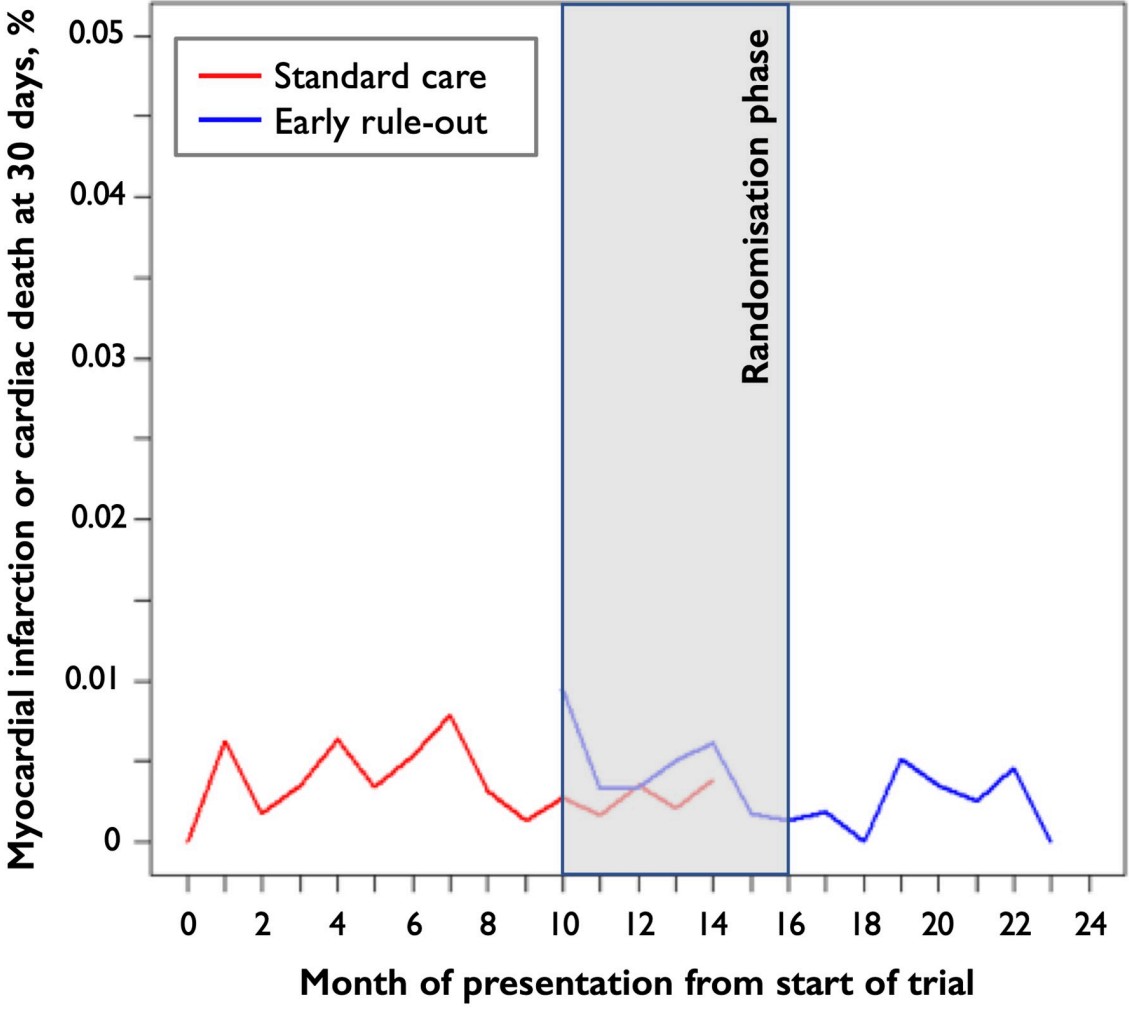

**Fig 4. Event rate for the primary safety outcome of myocardial infarction or cardiac death at 30 days by month of presentation from the start of the trial.**

It was informative to further examine the raw safety data after splitting the number of safety events (type 1 or type 4b myocardial infarction or cardiac death) by cluster-period. Here we let each cluster-period be defined as each site *and* month combination, so this means we split the number of safety events by site and by month.

After examining the raw safety endpoint data, we discovered that results were strongly driven by the safety outcome values of one or two sites under the intervention condition in the randomisation phase. Although it was not possible to export a full table of raw safety endpoint data to show in this article in case this identified individual patients, we are able to give some limited details below.

The median number of events per month at each site (cluster-period) in the randomisation phase was 1. However, at one site in one month during the randomisation phase, the number of events was notably higher (7 events) than in all other cluster periods, but still within the realms of random chance. It was instructive to assess the sensitivity of the results to the number of events in this cluster period. After reducing the number of events from 7 down to 1 (the median), and repeating the primary analysis, the odds ratio was reduced from 1.97 (95% CI 0.95 to 4.08) to 1.20 (95% CI 0.56 to 2.57). This is a substantial change after modifying the events from a single site during a single month of the trial.

We then investigated: "How many events do we need to subtract from those occurring under the intervention condition in the randomisation phase in order to reverse the direction of the odds ratio?" In the HiSTORIC trial, there was a 0.3% event rate under the intervention condition compared to a 0.4% event rate under the standard care condition. In contrast, when considering the randomisation phase only, we observed a 0.6% event rate (29 events out of 5248 presentations) under the intervention condition; compared to a 0.2% event rate (8 out of 4,088 presentations) under the control condition. If hypothetically the event rate under the intervention condition were 21 out of 5248, the results would have been: OR 1.00 (95% CI 0.46 to 2.16). Therefore, eight fewer events would have been needed to switch the direction of the odds ratio from 1.97 to just below 1.

Regarding possible exposure time effects, there was no clear evidence of changes in the event rate over time after implementation of the intervention within each site when examining the raw data.

## Discussion

### Results interpretation

Post-hoc results from extensive exploratory analyses of the HiSTORIC trial data are presented in this article. These results have confirmed the findings presented in the main trial paper [3]. In particular, we have confirmed the divergence between the calendar-matched and other analyses, even after utilising three distinct covariate adjustment methods.

Aside from the calendar-matched and "as-randomised" analyses, the length of stay outcome results were very similar regardless of the model specification or method of covariate adjustment. Although the calendar-matched analysis gave different point estimates and confidence intervals, our overall conclusion about the effectiveness of the intervention in reducing length of stay was unchanged. There was strong evidence of a reduction in length of stay based on the new pathway.

The binary safety endpoint results exhibited a similar pattern, except that results were less robust to the model specification and varied slightly more across the different methods of covariate adjustment. In general, the IPTW method provided more precise results (narrower confidence intervals) compared to the other methods, and regression adjustment for propensity score was not always equivalent to standard covariate adjustment.

There was a clear difference observed between the "as-randomised" and primary analysis results for both the efficacy and safety outcomes. Whilst all sites implemented during the 6-month randomisation phase, some implemented one or two months earlier and others later than planned due to local site pressures. Therefore the "as randomised" sensitivity analysis for the efficacy outcome measure is less helpful because the duration of stay will remain unchanged if the pathway has not actually been implemented, thereby diluting the intervention effect. Nevertheless, the magnitude of the difference observed between the primary analysis and the "as randomised" analysis for the safety outcome is potentially informative and suggested that the primary statistical model was overly influenced by data near the cross-over points or by the pattern of the cross-over points.

The raw safety outcome data showed no clear evidence of changes in the event rate over time after implementation of the intervention within each site. This appears to contradict the fact that after adjusting for exposure time in the regression models, the odds ratio point estimate (and 95% CI) reduced dramatically to be closer to 1. However, these results may be due to the cluster-period event count of 7 leading to a raised event rate in the randomisation phase and subsequent lower monthly event rates rather than any real exposure time effect. The event rate was only 0.2% in the third (implementation) phase, which was lower than the 0.6% event rate observed in the randomisation phase for patients recruited under the intervention condition. The exposure time model may therefore be overfitting the data.

Sparse data can in general lead to issues with model fitting and parameter estimates which are biased away from the null [21]. We explored whether a modified data augmentation method could be used to address this possible bias. However, the data augmentation method suggested that the effect of sparse data alone on the primary analysis model was insufficient on its own to explain the divergent results. The odds ratio point estimate and confidence intervals remained dissimilar to the calendar matched analysis results even after data augmentation.

## Evaluation of the study design

The HiSTORIC design was unusual in that it included a small number of sites and short duration of the randomisation phase. With a small number of sites, we are vulnerable to any confounding bias occurring within a single site, and this may have affected the results. Furthermore, the long before and after periods in the HiSTORIC trial meant that there was limited opportunity for contemporaneous comparisons as less than a quarter of the total trial duration covered periods in which sites were in both the control and intervention conditions. However, as the patient population is stable and no other changes in clinical practice for the assessment or treatment of patients without myocardial infarction were introduced during the study period it is unlikely that comparisons before and after the intervention was implemented are confounded by time varying trends.

## Evaluation of the statistical modelling approach

We performed a simulation analysis of the binary safety outcome (see S1 File) which showed no evidence of systematic bias in the intervention effect and no evidence of an inflated Type 1 error rate due to the very low number of clusters. This finding differs from Taljaard et al. [26], possibly because the between-cluster variability in our trial was effectively zero and the number of patients per site (cluster size) was so large. Furthermore, our choice of primary analysis model is supported by the literature; generalized linear mixed model (GLMM) analysis is the recommended approach for dealing with binary outcomes with a small number of clusters [27]. The use of Generalised Estimating Equations (GEE) with small sample correction has also been recommended in the literature [28, 29]. However, the models were either extremely

slow to converge or failed to converge completely in R software, even after fitting a very simple GEE model with exchangeable correlation structure (and no small sample correction). Our problems with fitting the GEE models were presumably due to the very large cluster sizes.

## Horizontal versus vertical bias

The avoidance of "horizontal bias" in stepped wedge trials in the form of adjusting for secular changes over time is strongly emphasised in the literature. In particular, under-specification of the secular trend will lead to bias in the intervention effect estimates [12]. For stepped wedge trials with high numbers of sites (i.e. over 50), bias across sites within cluster periods (i.e. "vertical bias") is less of a concern because any such bias is likely to have lower influence on the intervention effect estimate. However, we would suggest that "vertical bias" is likely to be of a much greater concern in stepped wedge trials of binary outcomes with a low number of clusters, particularly if the outcome event rate is low.

In the HISTORIC trial, we believe that the vertical comparisons in the randomisation phase and binary event rates near the cross-over points were highly influential for the primary analysis method. Our examination of the raw data supports this to some extent. Only eight fewer events were needed to switch the direction of the odds ratio from 2 to below 1 in the primary analysis method. It is possible that the reasons why the sites did not conform to the "as randomised" cross-over points affected the binary event rates near the cross-over points and introduced confounding bias. Indeed, this is perhaps likely as some sites implemented the early rule-out pathway early to address over-crowding in the Emergency Department at the start of the winter period.

Research by Kasza, Forbes and Taljaard [22, 23] suggests that the information content is not equal across cluster periods and that cluster-periods closest to the treatment switch date are most informative and contribute the most to the statistical analysis. Our concern was that the primary analysis model, although valid in the general sense, was too sensitive to any bias or random chance differences occurring near the crossover points. Specifically, our primary analysis model may have over-adjusted for unreal time effects in the short period where both the standard care and intervention conditions were running.

An advantage of the calendar matched before and after analysis is that it does not use vertical comparisons (which may be affected by confounding bias), and avoids the problem of attempting to model *both* the secular time trend and intervention effect based on a low event rate. We accept however the limitations of the calendar-matched analysis such as the potential for confounding by natural secular changes over time and reduced sample size. Nonetheless, the calendar-matched analysis was particularly suitable as a sensitivity analysis in the HiSTORIC trial due to the long before and after periods and small number of clusters in the design.

## Strengths

A strength of the study is that the analyses were based on a very large unselected population, with complete enumeration within each cluster over the study period and very little missing length of stay outcome data (0.05%). Identification and recruitment bias [30] was very unlikely in this trial because although clinicians at each site would have been aware of the control/intervention status within each site, the inclusion criteria was very broad (consisting of all patients presenting at emergency departments with suspected acute coronary syndrome) and the HiSTORIC screening tool was embedded into the patient record [3]. In addition, patients did not individually consent to the study and so there was virtually no prospect of the patients self-

selecting which intervention/control group to be recruited into, or indeed patients influencing outcomes due to knowledge of the intervention/control group.

## Limitations

The additional analyses presented in this article were all post-hoc and were conducted after the main HiSTORIC trial analyses. They therefore should be considered as purely exploratory and supportive of the findings shown in the main trial report [3]. We also report on a few experimental methods such as "data augmentation" and "adjustment for propensity score". Although adjustment for propensity score is a widely used method in clinical research, it has not to our knowledge been previously applied in the context of stepped wedge trials. Further research is needed to evaluate the performance of these methods in this context. In addition, the limitations mentioned in the main HiSTORIC trial report also carry forward into this study. In particular, the standard care arm of the HiSTORIC trial used a serial testing strategy based on the time of onset of symptoms, rather than a fixed time point 3 to 6 hours from presentation, which is more commonly used in other countries [3]. Furthermore, the statistical models were fitted using the lme4 package [9] in R software [24], which makes the assumption of a simple correlation structure of compound symmetry for linear mixed effects models (i.e. no autocorrelation in model residuals). Although we attempted to fit models assuming autoregressive or general (unstructured) correlation using the nlme package [31], the models would not fit, presumably for the same reason as for the GEE model: the within-cluster sizes were too large. Nevertheless, we believe that compound symmetry is a reasonable assumption given that independent individuals attending emergency departments within each site may be expected to be approximately equally correlated (or uncorrelated) with each other conditional on the fixed effects including season and time of presentation. Indeed, a few of the models fitted include sophisticated modelling of the time trend (e.g. via splines), which we think makes the compound symmetry assumption even more plausible because residual autocorrelation due to incomplete modelling of the time trend is less likely.

## Conclusions

After conducting in-depth post-hoc exploratory analyses of the HiSTORIC trial, we think a combination of issues jointly contributed to the divergent results: sparse event data, low number of sites, short randomisation phase, and non-adherence to the planned intervention schedule.

If a randomised trial is planned with a binary outcome and sparse event data is expected, then careful thought needs to be given at the study design stage as to whether a stepped wedge design is the most appropriate design. In stepped wedge trials, it is often necessary to use complex modelling to adjust for natural secular changes over time and adjust for between-cluster differences; but if the outcome is binary and the data sparse then this may cause difficulties with model fitting and interpretation. Stepped wedge trials of sparse binary data should not be considered lightly, particularly if the number of clusters available to be randomised is small. The final analysis of such trials should be supported by comprehensive sensitivity analyses and careful examination of the raw data.

Our analyses also raise concerns regarding the strong influence of data near the crossover points in stepped wedge trials with a small number of clusters when mixed effects models are applied. We recommend that future stepped wedge trials with a low number of expected clusters (i.e. below 20) should consider how vulnerable they are to "vertical bias" as well as "horizontal bias": the relative importance of each of these depends on the precise trial design.

Before-and-after analytical approaches that adjust for differences in patient populations but avoid direct modelling of the time trend should be considered.

## Supporting information

**S1 File. Simulation analysis.**
(PDF)

## Acknowledgments

We would like to thank Chris Tuck, Clinical Research Programme Manager at the Centre for Cardiovascular Science, for help with preparing this manuscript. We acknowledge the support of the National Health Service Research Scotland Lothian Research Safe Haven.

## Author Contributions

**Conceptualization:** Richard A. Parker, Nicholas L. Mills.

**Data curation:** Atul Anand.

**Formal analysis:** Richard A. Parker, Catriona Keerie.

**Investigation:** Richard A. Parker, Atul Anand, Nicholas L. Mills.

**Methodology:** Richard A. Parker, Catriona Keerie, Christopher J. Weir.

**Resources:** Atul Anand.

**Software:** Richard A. Parker.

**Supervision:** Christopher J. Weir, Nicholas L. Mills.

**Writing – original draft:** Richard A. Parker.

**Writing – review & editing:** Richard A. Parker, Catriona Keerie, Christopher J. Weir, Atul Anand, Nicholas L. Mills.

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
