## [Decision Letter · Decision Letter 0]

9 Feb 2022

PONE-D-21-36318Divergent confidence intervals among pre-specified analyses in the HiSTORIC stepped wedge trial: an exploratory post-hoc investigationPLOS ONE

Dear Dr. Parker,

Thank you for submitting your manuscript to PLOS ONE. After careful consideration, we feel that it has merit but does not fully meet PLOS ONE’s publication criteria as it currently stands. Therefore, we invite you to submit a revised version of the manuscript that addresses the points raised during the review process.

We look forward to receiving your revised manuscript.

Kind regards,

Salvatore De Rosa

Academic Editor

PLOS ONE

Journal Requirements:

Reviewers' comments:

Reviewer's Responses to Questions

**Comments to the Author**

1. Is the manuscript technically sound, and do the data support the conclusions?

Reviewer #1: Yes

Reviewer #2: Yes

2. Has the statistical analysis been performed appropriately and rigorously? 

Reviewer #1: Yes

Reviewer #2: Yes

3. Have the authors made all data underlying the findings in their manuscript fully available?

Reviewer #1: No

Reviewer #2: No

4. Is the manuscript presented in an intelligible fashion and written in standard English?

Reviewer #1: Yes

Reviewer #2: Yes

5. Review Comments to the Author

Reviewer #1: Several post-hoc statistical models were fitted for the primary outcomes of length of hospital stay and safety events. Models were adjusted for exposure time and incorporated spline fitting and a random effect for the time effect. The methods used were direct inclusion, regression adjustment for propensity score, and weighting, as well as a data augmentation approach. The new statistical models confirmed the results of the pre-specified trial analysis where the event rate was low: 0.36%.

Minor revision:

Table 1: In the “Primary analysis” specify the type of statistical model used for the analysis.

Reviewer #2: Parker and colleagues should be congratulated for their efforts carrying out this important analysis. In their current paper, they investigate in very carefully robust analysis the reasons for the divergent results that were reported in the HiSTORIC trial. The current paper helps understand the results of the HiSTORIC trial more, and can assist researches with the design and implementation of future stepped-wedge cluster randomised trials. Parker and colleagues have confirmed the results of the pre-specified trial analysis and have validate their complexity. Several major comments:

1. Although the design and the execution of the current analysis are very elegant, the additive scientific value is not well presented. The conclusion at the end of the manuscript is very long and should be shortened and more focused.

2. Overall the discussion and the conclusion are very long and it's hard to follow. Would highly recommend to be more focused.

3. In the discussion the authors discuss several potential reasons for the differences between certain results under

separate headings. I would recommend to choose two or three that the authors believe are the most important ones, expand on them, and mention the rest briefly. It's difficult for a non-statistician to understand which reason affected the diversity the most.

Minor comments:

1. Limitation. Each study including the current one is prone to several limitations. I would highly recommend mention the limitation of the current study: post hoc; selective population etc..

2. Figure labels for the ratio and the CI should be added.

3. The Results section contains many explanations for the findings. These should be moved to the discussion. The Result section should report findings only and not their interpretations or explanations.

6. PLOS authors have the option to publish the peer review history of their article (what does this mean?). If published, this will include your full peer review and any attached files.

Reviewer #1: No

Reviewer #2: **Yes: **Arwa Younis

---

## [Author Response · Author response to Decision Letter 0]

3 Mar 2022

We would like to thank the associate editor and reviewers for their constructive comments, which we are pleased to address in the revised version of the manuscript. We give a point-by-point response to the associate editor and reviewers’ comments below:

Journal Requirements:

Thank you for your comment. We have now modified the manuscript so that it meets the PLOS ONE style requirements.

Participant consent was not sought for our study and this was approved by the NHS Scotland Public Benefit And Privacy Panel For Health And Social Care (NHSS HSC-PBPP) committee. We have now clarified this in the methods section (page 5): “We received approval to use routinely collected National Health Service (NHS) data for research without the need for individual patient consent. This approval was granted by the NHS National Services Scotland Privacy Advisory Committee (PAC), now called the NHS Scotland Public Benefit And Privacy Panel For Health And Social Care (NHSS HSC-PBPP).” 

Reviewers' comments:

Reviewer #1: Several post-hoc statistical models were fitted for the primary outcomes of length of hospital stay and safety events. Models were adjusted for exposure time and incorporated spline fitting and a random effect for the time effect. The methods used were direct inclusion, regression adjustment for propensity score, and weighting, as well as a data augmentation approach. The new statistical models confirmed the results of the pre-specified trial analysis where the event rate was low: 0.36%.

Minor revision:

Table 1: In the “Primary analysis” specify the type of statistical model used for the analysis.

Thank you for your comment. We have now specified that we used mixed effects models in the first row (below the headings) of Table 1. We have also clarified this in Table 2. 

Reviewer #2: Parker and colleagues should be congratulated for their efforts carrying out this important analysis. In their current paper, they investigate in very carefully robust analysis the reasons for the divergent results that were reported in the HiSTORIC trial. The current paper helps understand the results of the HiSTORIC trial more, and can assist researches with the design and implementation of future stepped-wedge cluster randomised trials. Parker and colleagues have confirmed the results of the pre-specified trial analysis and have validate their complexity. 

Thank you very much for these comments!

Several major comments:

1. Although the design and the execution of the current analysis are very elegant, the additive scientific value is not well presented. The conclusion at the end of the manuscript is very long and should be shortened and more focused.

Thank you for your comment. We have now made the conclusions section more focussed and have reduced its length by 117 words.

2. Overall the discussion and the conclusion are very long and it's hard to follow. Would highly recommend to be more focused.

Thank you for your comment. We agree, and have removed a substantial amount of material from the discussion section, including our long list of recommendations, which we felt was overly repetitive and not very focussed. As a whole, the length of these sections has reduced by almost 400 words, despite us adding in more material in response to your other helpful comments regarding including “explanations from the results section” and also including a strengths and limitations sections. We have made substantial changes to the discussion and conclusions sections so that they are now more focussed and easier to read.

3. In the discussion the authors discuss several potential reasons for the differences between certain results under separate headings. I would recommend to choose two or three that the authors believe are the most important ones, expand on them, and mention the rest briefly. It's difficult for a non-statistician to understand which reason affected the diversity the most.

Our changes (mentioned above in response to point 2) have helped to address these concerns and have made the discussion section more focussed. In particular, we have reduced the number of headings we are using in the discussion section. Our main conclusions regarding the reason for differences between results are highlighted in the Conclusions section (as well as covered in the section entitled “Horizontal versus vertical bias”). In the conclusions section we write: “we think a combination of issues jointly contributed to the divergent results: sparse event data, low number of sites, short randomisation phase, and non-adherence to the planned intervention schedule.”, and then go on to expand briefly on the key issues of sparse event data (in combination with low number of sites) and strong influence of data near the crossover points. 

Minor comments:

1. Limitation. Each study including the current one is prone to several limitations. I would highly recommend mention the limitation of the current study: post hoc; selective population etc..

Thank you for your comment. We agree, and have now added a limitations section to the Discussion. In addition, we feel that the HISTORIC trial had important strengths that we need to acknowledge, and so we have also mentioned these in a “strengths” section.

2. Figure labels for the ratio and the CI should be added.

Thank you for your comment. We have now added the clarifying information (“geometric mean ratio” and “95% confidence intervals”) to the figure labels. 

3. The Results section contains many explanations for the findings. These should be moved to the discussion. The Result section should report findings only and not their interpretations or explanations.

Thank you for your comment. We have moved explanations for the findings contained in the results section to the Discussion section (particularly the last two paragraphs of the results section).

---

## [Decision Letter · Decision Letter 1]

6 Apr 2022

PONE-D-21-36318R1Divergent confidence intervals among pre-specified analyses in the HiSTORIC stepped wedge trial: an exploratory post-hoc investigationPLOS ONE

Dear Dr. Parker,

Thank you for submitting your manuscript to PLOS ONE. After careful consideration, we feel that it has merit but does not fully meet PLOS ONE’s publication criteria as it currently stands. Therefore, we invite you to submit a revised version of the manuscript that addresses the points raised during the review process.

In particular, the external Reviewer #1 has additional minor requests. Please, make sure to address all residual criticism.

We look forward to receiving your revised manuscript.

Kind regards,

Salvatore De Rosa

Academic Editor

PLOS ONE

Journal Requirements:

Reviewers' comments:

Reviewer's Responses to Questions

**Comments to the Author**

1. If the authors have adequately addressed your comments raised in a previous round of review and you feel that this manuscript is now acceptable for publication, you may indicate that here to bypass the “Comments to the Author” section, enter your conflict of interest statement in the “Confidential to Editor” section, and submit your "Accept" recommendation.

Reviewer #1: (No Response)

2. Is the manuscript technically sound, and do the data support the conclusions?

Reviewer #1: Yes

3. Has the statistical analysis been performed appropriately and rigorously? 

Reviewer #1: Yes

4. Have the authors made all data underlying the findings in their manuscript fully available?

Reviewer #1: No

5. Is the manuscript presented in an intelligible fashion and written in standard English?

Reviewer #1: Yes

6. Review Comments to the Author

Reviewer #1: The results and conclusions from a relatively new study design are now more succinctly and clearly summarized.

Minor Revision:

State the underlying covariance structures used in the mixed effects models and the criteria for selecting them.

7. PLOS authors have the option to publish the peer review history of their article (what does this mean?). If published, this will include your full peer review and any attached files.

Reviewer #1: No

---

## [Author Response · Author response to Decision Letter 1]

24 May 2022

We would like to thank the reviewer for their comment, which we are pleased to address in the revised version of the manuscript. 

Reviewer #1: The results and conclusions from a relatively new study design are now more succinctly and clearly summarized.

Minor Revision:

State the underlying covariance structures used in the mixed effects models and the criteria for selecting them.

Thank you for your comment. We have now specified in the paper that a “compound symmetry variance-covariance structure was assumed” for the linear mixed effects models (page 6).

We have also added a limitation to the limitations section of the discussion section (pages 17-18): 

“Furthermore, the statistical models were fitted using the lme4 package [9] in R software [24], which makes the assumption of a simple correlation structure of compound symmetry for linear mixed effects models (i.e. no autocorrelation in model residuals). Although we attempted to fit models assuming autoregressive or general (unstructured) correlation using the nlme package [31], the models would not fit, presumably for the same reason as for the GEE model: the within-cluster sizes were too large. Nevertheless, we believe that compound symmetry is a reasonable assumption given that independent individuals attending emergency departments within each site may be expected to be approximately equally correlated (or uncorrelated) with each other conditional on the fixed effects including season and time of presentation. Indeed, a few of the models fitted include sophisticated modelling of the time trend (e.g. via splines), which we think makes the compound symmetry assumption even more plausible because residual autocorrelation due to incomplete modelling of the time trend is less likely.”

We confirm that a reference to the “nlme” package has now been added to our reference list (number [31]).

---

## [Decision Letter · Decision Letter 2]

23 Jun 2022

Divergent confidence intervals among pre-specified analyses in the HiSTORIC stepped wedge trial: an exploratory post-hoc investigation

PONE-D-21-36318R2

Dear Dr. Parker,

We’re pleased to inform you that your manuscript has been judged scientifically suitable for publication and will be formally accepted for publication once it meets all outstanding technical requirements.

Kind regards,

Salvatore De Rosa

Academic Editor

PLOS ONE

**Comments to the Author**

1. If the authors have adequately addressed your comments raised in a previous round of review and you feel that this manuscript is now acceptable for publication, you may indicate that here to bypass the “Comments to the Author” section, enter your conflict of interest statement in the “Confidential to Editor” section, and submit your "Accept" recommendation.

Reviewer #1: All comments have been addressed

2. Is the manuscript technically sound, and do the data support the conclusions?

Reviewer #1: (No Response)

3. Has the statistical analysis been performed appropriately and rigorously? 

Reviewer #1: (No Response)

4. Have the authors made all data underlying the findings in their manuscript fully available?

Reviewer #1: (No Response)

5. Is the manuscript presented in an intelligible fashion and written in standard English?

Reviewer #1: (No Response)

6. Review Comments to the Author

Reviewer #1: (No Response)

7. PLOS authors have the option to publish the peer review history of their article (what does this mean?). If published, this will include your full peer review and any attached files.

Reviewer #1: No

---

## [Editor Report · Acceptance letter]

24 Jun 2022

PONE-D-21-36318R2 

Divergent confidence intervals among pre-specified analyses in the HiSTORIC stepped wedge trial: an exploratory post-hoc investigation 

Dear Dr. Parker:

I'm pleased to inform you that your manuscript has been deemed suitable for publication in PLOS ONE. Congratulations! Your manuscript is now with our production department. 

Kind regards, 

on behalf of

Dr. Salvatore De Rosa 

Academic Editor

PLOS ONE